# Electrodeposition of Nanostructured Co–Cu Thin Alloy Films on to Steel Substrate from an Environmentally Friendly Novel Lactate Bath under Different Operating Conditions

Raiedhah A. Alsaiari [1], Medhat M. Kamel [2,*] and Mervate M. Mohamed [1,2]

1  Empty Quarter Research Centre, Department of Chemistry, Faculty of Science and Arts in Sharurah, Najran University, Sharurah 78362, Saudi Arabia; raalsayari@nu.edu.sa (R.A.A.); mervate71@gmail.com (M.M.M.)
2  Department of Chemistry, Faculty of Science, Suez Canal University, Ismailia 41522, Egypt
*  Correspondence: medhat_darwish@science.suez.edu.eg; Tel.: +20-100-448-2037; Fax: +20-64-323-0416

**Abstract:** A new lactate bath was proposed to deposit Co–Cu thin alloy films in nanostructure form onto a steel cathode. The deposition bath contained $CuSO_4.5H_2O$, $CoSO_4.7H_2O$, $CH_3CHOHCOOH$, and anhydrous $Na_2SO_4$ at pH 10. The effects of $[Co^{2+}]/[Cu^{2+}]$ molar ratios, lactate ion concentration, current density (CD), and bath temperature on cathodic polarization, cathodic current efficacy (CCE), composition, and structure of the Co–Cu alloys were investigated. The new bath had a high cathodic current efficiency of 85%, which increased with the applied CD. However, it decreased as the temperature increased. The produced coatings have an atomic percentage of Cu ranging from 19.8 to 99%. The deposition of the Co–Cu alloy belonged to regular codeposition. The Co content of the deposit increased with the amount of $Co^{2+}$ ions in the bath, lactate concentration, and current density but decreased as the temperature increased. Cobalt hexagonal close-packed (HCP) and copper-rich, face-centered cubic (FCC) Co–Cu phases combine to form the polycrystalline structure of the electrodeposited Co–Cu alloy. The average crystallite size ranges between 46 and 89 nm. An energy dispersive X-ray (EDX) examination confirmed that the deposit contained Cu and Co metals. The throwing power and throwing index of the alkaline lactate bath were evaluated and found to be satisfactory.

**Keywords:** electrodeposition; Co–Cu; steel; lactate bath; thin films; nanostructures





## 1. Introduction

Magnetic media, computers, and the microelectronics sector all make extensive use of amorphous ferromagnetic films. These films are prepared using a variety of techniques. A good method for creating highly functional magnetic recording materials is electrodeposition [1,2]. It offers the advantage of low-cost production since it requires simple and inexpensive processing equipment. Additionally, the electrodeposition process can be used to create multilayers of any shape and size [3–7].

The electrodeposition of Co–Cu alloys on a substrate surface is of great interest as they are used in several different industrial processes. Depending on the amount of Co in the alloy and the substrate used, Co–Cu coatings have a variety of fascinating applications. For example, alloys with low cobalt content exhibit high magnetoresistance. It can be utilized in sensor technologies and data storage systems when deposited on silicon, copper, or platinum substrates [8–10]. In addition, the fabrication of Co–Cu alloy coatings with increased cobalt concentrations on other substrates can be utilized for catalytic processes [9,11–13] and anticorrosive coatings [14–18].

Co–Cu alloy coatings are usually prepared using physical deposition methods [19,20]. The formation of the coatings using the electrodeposition technique provides a potential substitute for enhancing the effectiveness of the deposition process and decreasing cost. However, because the reduction potentials of Cu (II) and Co (II) ions differ by −0.60 V,

it is impossible to simultaneously reduce both cations on the cathode without the use of a complexing agent [15–17]. Cyanide ion has long been the most used ligand for the electrodeposition of copper alloys, despite its toxicity [17,18,21]. To solve this problem, alloy coatings can be manufactured using environmentally benign compounds as complexing agents [12,22–25].

It may be possible to lower the costs of the electroplating industry by employing less hazardous electrolytes, which are closely related to the treatment of effluents and meet the requirements for reinforced exhaust equipment.

Direct current (DC) or pulsed current (PC) can be used to deposit Co–Cu alloy coatings. Each method has a different impact on the electrical double layer, current distribution, and mass transport, resulting in coatings with a variety of compositions, morphologies, and surface roughness [26–32]. Although the use of DC is less expensive and simpler, studies suggest that using pulsed current has certain benefits for alloy electrodeposition, such as improving the adhesion, density, resistivity, and ductility of coatings. In addition, it is possible to form layers with fine grains, greater morphological consistency, and reduced porosity [25,33–35].

In several instances, the selection of an electrolyte that has the correct composition is significant for obtaining alloys with the composition and structure needed for usage. Evidence suggests that Co–Cu alloys can be electrodeposited in a variety of electrolytes, such as sulphate [36–38], citrate [39–42], or glycinate [23]. Additionally, it has been discovered that various organic additives, such as dodecyl sulphate, saccharin, and glycine, have an impact on the magnetic characteristics of Co–Cu alloys and multiple layers in addition to their structural effects [43].

Hagiwara et al. [44] produced a Co–Cu alloy film with a high composition gradient using a novel electrodeposition technique that involves continually varying the applied voltage. The composition–gradient Co–Cu alloy film's hardness was assessed. The hardness values were compared by the authors with those of Co–Cu alloy films formed in a multilayered Co–Cu film and under constant potentials. Five distinct elements-Ni, Co, Cu, Mo, and W-were deposited as high-entropy alloy coatings by Ahmadkhaniha et al. [45] using a single aqueous bath. The impacts of pH, current density, and the complexing agent on the coating composition were investigated. The results indicated that Mo and W were preferentially codeposited with Ni and Co. The anticorrosive characteristics of CuCoNi, CuCo, and CuNi alloy films electrodeposited via DC were assessed by de Souza et al. [46] in a NaCl 0.5 mol $L^{-1}$ solution. The findings demonstrated that the current density and the bath composition impacted the chemical, morphological, and electrochemical features of the alloy films.

On the previously prepped aluminum substrates, Kazadi et al. [47] electrodeposited a series of Co–Cu granular thin film alloys at different deposition times. The structure and magneto-optical characteristics were investigated in detail. The XRD results showed that a mixture of face-centered cubic (fcc) and hexagonal close-packed (hcp) phases existed in all films. El-Tahawy et al. [48] electrodeposited Co-rich Co–Cu and Co–Ni alloys by adding different quantities of $CuSO_4$ and $NiSO_4$, respectively, to the $CoSO_4$ bath. They found that, up to roughly 2 at% of the alloying element, only the hcp phase formed in the Co–Cu and Co–Ni systems. A notable fcc phase fraction was observed in Co–Cu above this concentration, while a smaller fcc fraction was observed in Co–Ni up to close to 8 at%.

As the conjugate base of lactic acid, lactate is a hydroxy monocarboxylic acid anion, created via the deprotonation of the carboxylic group. Lactate ions can be employed as a substitute complexing agent for the electrodeposition of metal coatings. When combined with Cu (II) and other metal (II) ions, lactate anions form stable complexes [49–52].

The current study aims to investigate the electrodeposition of Co–Cu alloys on a low-carbon steel substrate from a novel bath containing lactate ions as a ligand, using a DC source. The high solubility of lactate ions in the plating bath led to their selection. It does not harm the environment and is readily available and inexpensive. The impacts of the bath composition ($[Co^{2+}]/[Cu^{2+}]$ molar ratio and lactic acid concentration) and

operating parameters (current density and temperature) on the cathodic polarization, cathodic current efficiency, deposited film composition, structure, morphology, throwing power, and throwing index of the bath were investigated. For the forthcoming paper, the corrosion resistance of the Co–Cu alloy obtained from lactate bath will be examined in acidic environments using different techniques.

## 2. Materials and Methods

In all experiments, the cathodes were made from steel sheets (0.5 mm thickness) of the following composition (wt.%): C, 0.08; Si, 0.01; Mn, 0.030; P, 0.025; S, 0.025; and Al, 0.045%. The steel sheets were produced locally by the Egyptian Iron and Steel Company (Helwan, Egypt). Prior to measurement, the cathodes were subjected to degreasing and polishing treatments as follows: degreasing was carried out by immersing the cathodes in petroleum ether (60–80 °C) for 24 h, then they were washed with water and dried [42].

The anodes were made from pure platinum metal. They were cleaned by dipping them in concentrated hydrochloric acid for 30 seconds, then washed with running water and distilled water, and finally rinsed with a small amount of the electrolyte under study.

Electroplating of the Co–Cu alloy was conducted under a constant current (3.33–11.55 mA cm$^{-2}$) from a solution containing $CoSO_4 \cdot 7H_2O$ (0.09–0.14 M), $CuSO_4 \cdot 5H_2O$ (0.01–0.06 M), anhydrous $Na_2SO_4$ (0.1 M), and lactic acid (0.5–1.0 M). The total concentration of metal sulphates was kept constant (0.15 M) in the plating bath. All solutions used in the present work were prepared using fine chemicals and distilled water. The solutions were freshly prepared prior to use. The pH of each solution was adjusted with ammonium hydroxide or sulfuric acid and measured using a Fisher Scientific pH meter Model 915 (San Diego, CA, USA).

The experimental setup consisted of a Perspex cell (0.06 L) in the shape of a rectangle with a Pt sheet anode and a steel sheet cathode. Each electrode had dimensions of $3 \times 3$ cm$^2$. After mechanically polishing the cathode with fine emery paper, it was cleaned with distilled water, washed with ethyl alcohol, and weighed. Stagnant solutions were used for the deposition process. The deposition time was 20 min; after that, the steel cathode was removed, rinsed with distilled water, left to dry, and weighed.

An atomic absorption spectrometer of Thermo S4Thermo SOLAAR (Manasquan, NJ, USA) was used to determine the alloy composition after the coatings were digested in 32% HCl and diluted with bi-distilled water. The CCE for Co and Cu electrodeposition was calculated by measuring the change in weight ($\Delta w$) of the deposit on the steel cathode.

$$\text{CCE} = \left(\frac{\Delta w}{w_t}\right) \times 100; \tag{1}$$

$$w_t = \frac{i \times t \times M}{n \times F} \tag{2}$$

where $\Delta w$ is the change in weight after electrodeposition (g), $w_t$ is the theoretical weight of the metal (g), $i$ is the applied current (A), $t$ is the deposition time (sec), $n$ represents the number of electrons transferred, $M$ is the molecular mass of metal, and $F$ is Faraday's constant (96,485 C/mol) [53].

The cathodic current efficiency of alloy deposition, $CCE_{alloy}$, equals the sum of the cathodic current efficiencies of the parent metals, Co and Cu.

$$CCE_{alloy} = CCE_{Co} + CCE_{Cu} \tag{3}$$

The percentage of Co in the alloy was determined using Equation (4).

$$Co(alloy)\% = \left(\frac{mass\ of\ Co\ (alloy)}{(mass\ of\ Co\ (alloy) + mass\ of\ Cu\ (alloy))}\right) \times 100 \tag{4}$$

Equation (5) was used to estimate the percentage of Co in the bath [54].

$$Co\ (bath)\ \% = \left( \frac{mass\ of\ Co\ (bath)}{(mass\ of\ Co\ (bath) + mass\ of\ Cu\ (bath))} \right) \times 100 \tag{5}$$

A Biologic SP-150 potentiostat/galvanostat (Bio-Logic USA, LLC, Knoxville, TN, USA) was utilized to carry out the polarization measurements. A three-electrode system was used in a traditional electrolytic jacketed glass cell (0.1 L): platinum gauze was used as the counter electrode, steel served as the working electrode, and SCE worked as the reference electrode (connected to the electrochemical system by a Luggin capillary). The steel electrode was cut into a cylindrical shape and covered with epoxy resin, allowing a 0.785 cm$^2$ surface area to contact the solution. Prior to conducting any experiments, the steel electrode was abraded using emery paper ranging in grade from 250 to 1200 and then cleaned using distilled water and acetone. The potentiodynamic polarization curves were swept from $-0.2$ to $-1.6$ V (SCE) with a scan rate of 20 mV s$^{-1}$.

The Haring and Blum cell, described previously [55], was used for throwing power (T.P) and throwing index (T.I) experiments. The plating cell was made from transparent Perspex in the form of a rectangular trough with inside dimensions of 20 cm length, 3 cm width, and 2.5 cm height. The cell was provided with one platinum anode between two parallel cathodes at different distance ratios (1:1–1:3).

The cell was cleaned and filled with plating solution (0.1 L). The electrodes were then inserted at appropriate positions at a distance ratio of 1:3. The two electrodes (near and far cathodes) were weighed before and after electrodeposition for a given time and current density. The metal distribution ratio (M), which is the ratio between the weights of the metal deposited at the near cathode to that at the far cathode, was calculated. The empirical Fields formula [56] was used to calculate the T.P at the linear distance ratio (L = 3).

$$T.P\% = \frac{L - M}{L + M - 2} \times 100 \tag{6}$$

For measurements of the T.I, the two cathodes were placed at different distance ratios from the anode (1:1, 1:2, and 1:3). The metal distribution ratio (M) was computed and plotted against the linear ratio (L). The linear ratio represents the current distribution ratio of the near cathode to the far cathode. The reciprocal of the slope of the plot (T.I) was used as an indicator for the T.P of the bath.

The phase and crystal structure of the deposits were examined using an X-ray diffractometer (PANalytical X-PERT PRO) (Massachusetts Institute of Technology, Sant Louis, MO, USA) with an iron filter and copper radiation, Cu-K$_\alpha$ ($\lambda$ = 1.54 °A). The X-ray tube was energized at 45 kV and 40 mA. The patterns of the tested specimens were recorded automatically with a scanning speed of 2 °C per minute. The diffraction patterns were recorded at room temperature.

The grain sizes (D) were estimated from the broadening of the reflections using Scherrer's formula [52]:

$$D = \frac{0.9\ \lambda}{\beta\ cos\ \theta} \tag{7}$$

where $\lambda$ is the X-ray wavelength (0.1540 nm), $\beta$ is the full-width-of-half maximum (FWHM), and $\theta$ is the reflection angle. The JOEL JSM-6510 LV low-vacuum SEM was operated at 30 kV to check the morphology of the coatings. Prior to examination, the samples were coated with a thin layer of gold to enhance the conductivity. The EDX technique (EDX-FEI-QUANTA FEG 250) was operated at 10 kV to identify the elemental composition of the coating. The EDX analysis was obtained as an average value of three points at the surface.

### 3. Results and Discussion

*3.1. Thermodynamics Study*

The Nernst equation is an important equation that relates the electrode potential (*E*) to the standard electrode potential (*E°*), the temperature (*T*), and the concentrations of the reactants and products of the half-cell reaction. By applying the Nernst equation for the reduction of $Cu^{2+}$ and $Co^{2+}$ ions from aqueous solutions containing their simple salts, we obtain the following:

$$E(Cu/Cu^{2+}) = 0.0985 + \frac{0.059}{2} log[Cu^{2+}]; \tag{8}$$

$$E(Co/Co^{2+}) = -0.5185 + \frac{0.059}{2} log[Co^{2+}] \tag{9}$$

where 0.0985 and −0.5185 are the standard reduction potentials in volts for copper and cobalt metals versus the saturated calomel electrode (SCE), respectively. Under the given experimental conditions, the concentration of $Cu^{2+}$ ions is 0.03 mol $L^{-1}$; however, the concentration of $Co^{2+}$ ions is 0.12 mol $L^{-1}$. The calculated deposition potentials are 0.0536 and −0.546 V for copper and cobalt, respectively. In practice, the potential required to achieve electrodeposition is typically higher than the calculated potential due to overpotential losses [57]. The losses are caused by kinetic factors such as the activation energy of the electrode reaction and the concentration gradients of the reactants and products at the electrode surface.

The most important factor in alloy electrodeposition is the difference in the metal ion species' deposition potentials. Considering the standard electrode potentials *E°*, which are 0.0985 V (SCE) for the $Cu/Cu^{2+}$ and −0.5185 V (SCE) for the $Co/Co^{2+}$ redox couple, this may provide a first estimation of their deposition behavior. It is clear from the *E°* values that more conditioning is required to achieve cobalt and copper codeposition because the deposition potentials must be like to facilitate alloy electrodeposition. According to the Nernst equation, as the metal ion activity in the electrolyte solution decreases, the reversible electrode potential (*E*) can be adjusted to increasingly negative values. Accordingly, the electrolyte bath's $Co^{2+}$: $Cu^{2+}$ ratio was adjusted to 4:1 to take advantage of the low copper concentration of 0.03 mol $L^{-1}$ and move the reversible electrode potentials of copper and cobalt closer together. However, this shift is not substantial enough to affect the codeposition system in a meaningful way. The most frequent and simple technique to reduce the apparent concentrations and bring the deposition potentials closer together is to add a complexing agent to the solution. In this work, lactate ion was utilized as a complexing agent [58,59].

When a lactate ion (*L*) is introduced into the plating solution at pH 10, new equilibriums will be established as follows [60,61]:

$$Cu(H_{-1}L) L^- + OH^- \rightleftharpoons Cu(H_{-1}L)_2^{2-} + H_2O \; (7.80 < pH < 13.15); \tag{10}$$

$$Co^{2+} + 2L + 2H_2O \rightleftharpoons [Co (L)_2(H_2O)_2]. \tag{11}$$

Using the formation constants of the formed complexes, the deposition potential of copper and cobalt can be calculated according to Equations (12) and (13) [59].

$$E \, (Cu/Cu^{2+}) = 0.0985 - \frac{0.059}{2} log \frac{[H_2O][Cu(H_{-1}L)_2]^{2-}}{[Cu(H_{-1}L)L]^-[OH]^-} \tag{12}$$

$$E \, (Co/Co^{2+}) = -0.5185 - \frac{0.059}{2} log \frac{[Co(L)_2(H_2O)_2]}{[Co^{2+}][L]^2[H_2O]^2} \tag{13}$$

The formation constant of the $[Cu(H_{-1}L)_2]^{2-}$ complex is $10^{5.05}$ [60] and that of $[Co(L)_2(H_2O)_2]$ is 6.456 [62]. By introducing the concentration of the given species in Equation (10), the deposition potential of copper is found to be −0.050 V; however, the

deposition potential of cobalt is $-0.542$ V. Therefore, even though the numerical calculations of the potentials of the copper and cobalt lactate complexes do not demonstrate a shift to more comparable potentials. Kinetic effects are believed to make an important contribution, enabling Co–Cu codeposition [57]. For the system employed here, Cu has a larger positive reversible potential than Co; therefore, it is more noble than Co and Cu is preferentially deposited, if at comparable bulk concentrations, than Co. Obtaining a cobalt-rich alloy might be challenging due to the difference in nobility. To decrease the content of Cu in a Co alloy, a mass transport limit of Cu is created by lowering its bulk concentration relative to Co. To deposit copper, the applied current density must remain below the copper-limiting current density. When the applied current density becomes bigger than the Cu-limiting current density, or the working electrode potential is larger than the Co equilibrium potential, the less noble Co begins to be deposited [63]. But, at the same time, copper is also deposited in the cobalt-layer. Because the copper concentration is low in the electrolyte, copper reduction is under transport control; thus, the copper content in the alloy is small and decreases when increasing the applied current density or decreasing temperature. The experimental conditions, such as electrolyte concentration, pH, and applied potential, can also influence the relative deposition rates of copper and cobalt.

### 3.2. Potentiodynamic Polarization Measurements

The cathodic polarization graphs for the electrodeposition of Co–Cu alloy onto a steel substrate under distinct testing parameters are shown in Figures 1–4. The curves were swept from $-0.2$ to $-1.6$ V (SCE) with a scan rate of 20 mV s$^{-1}$. The graphs display the relation between the recorded current density and applied cathodic potentials for the electrodeposition of Co, Cu, and Co–Cu alloys under the same conditions. According to Figure 1, complex formation causes a significant amount of polarization to accompany the discharge of the individual metals, Cu and Co. Along with other complex species, Cu was primarily found as the [Cu $(C_3H_5O_3)_2$] complex. Co was commonly present as the [Co $(C_3H_5O_3)_2$] complex [47]. At lower negative potentials ($<-1.13$ V), Cu was more noble than Co, as shown by the fact that the Co polarization graph has significantly more negative potentials than that of Cu. However, crossing of the Cu and Co curves was observed at a more negative potential ($>-1.13$ V). Co becomes a metal that is easier to deposit than Cu. Therefore, at the less negative potential, Cu is preferentially deposited, and Cu-rich alloys are formed [42]. Under these conditions, codeposition proceeds via Cu diffusion control. Lactic acid acts as a homogeneous catalyst for the reduction of Co$^{2+}$ ions at high negative potentials and codeposition proceeds via kinetic control [42,64]. The polarization curve of the Co–Cu alloy was in the middle of that of the parent metals. The codeposition enables the more noble metal (Cu) to discharge at more negative potentials and the less noble metal (Co) to discharge at more positive potentials [23].

The current complex solution commonly exhibits concentration polarization as Co and Cu are present in the solution as lactate complexes. It is possible to conduct the electroreduction processes mentioned below, with hydrogen evolution occurring because of a different side reaction [65].

$$[Cu(H_{-1}L)_2]^{2-} + 2e \rightleftarrows Cu_{(s)} + 2H_{-1}L^{-2} \tag{14}$$

$$[Co(L)_2(H_2O)_2] + 2e \rightleftarrows Co_{(s)} + 2L + 2H_2O \tag{15}$$

As Co$^{2+}$ and Cu$^{2+}$ ions are discharged, the amount of lactate ligand increases at the cathode boundary, altering the cathodic polarization towards higher negative values.

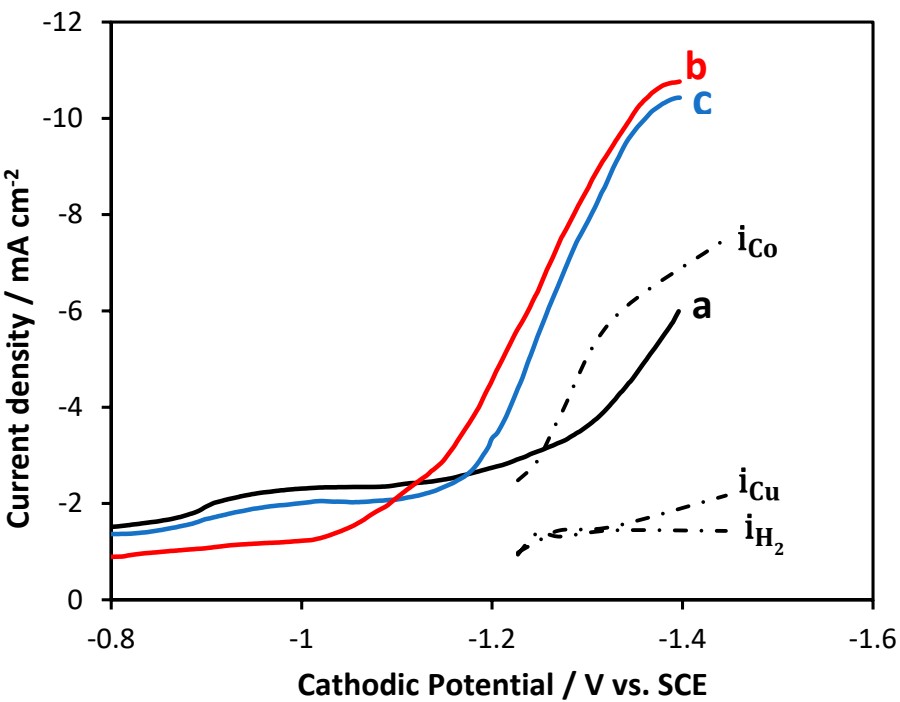

**Figure 1.** Cathodic polarization curves obtained at temperature = 25 °C and pH = 10 for the electrodeposition of the following: (**a**) Cu from solution containing 0.03 M $CuSO_4.5H_2O$; (**b**) Co from solution containing 0.12 M $CoSO_4.7H_2O$; (**c**) Co–Cu alloy from solution containing 0.12 M $CoSO_4.7H_2O$ and 0.03 M $CuSO_4.5H_2O$. The cathodic working electrode is steel. Each solution contains 0.5 M lactic acid and 0.1 M $Na_2SO_4$. (- - - -) Partial current densities of Co, Cu, and $H_2$ during the electrodeposition of Co–Cu alloy at a scan rate of 20 mV/s.

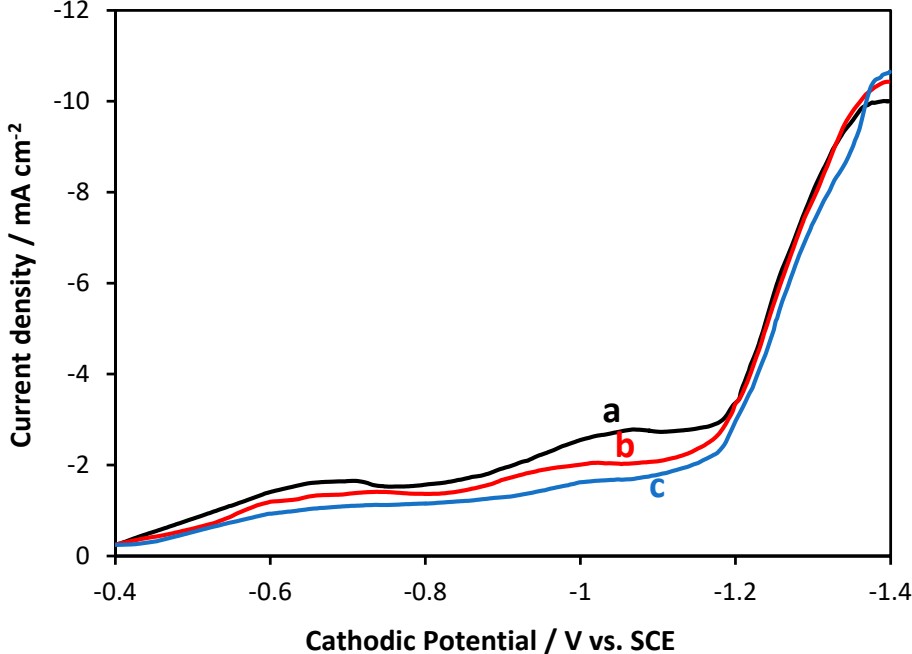

**Figure 2.** Cathodic polarization curves for the electrodeposition of Co–Cu alloy onto steel cathodes at pH = 10 and temperature = 25 °C from solutions containing 0.5 M lactic acid and 0.1 M $Na_2SO_4$ and different molar ratios of $[Co^{2+}]/[Cu^{2+}]$: (**a**) 2; (**b**) 4; (**c**) 6.5. Scan rate = 20 mV/s.

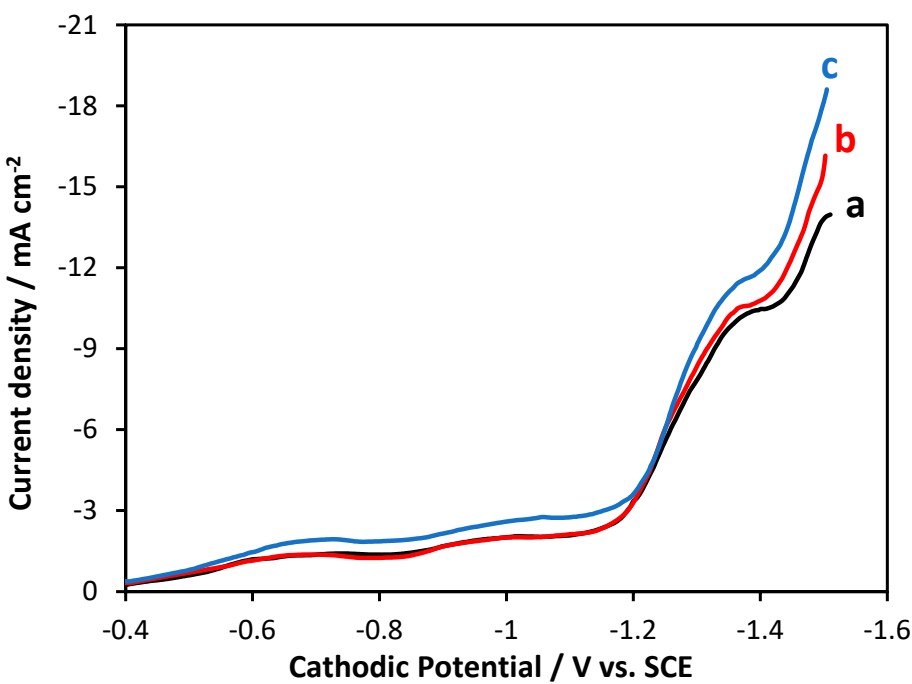

**Figure 3.** Cathodic polarization curves for the electrodeposition of Co–Cu alloy onto steel cathodes at pH = 10 and 25 °C from solutions containing 0.12 M $CoSO_4.7H_2O$, 0.03 M $CuSO_4.5H_2O$, and 0.1 M $Na_2SO_4$ and different concentrations of lactic acid: (**a**) 0.5; (**b**) 0.6; (**c**) 0.8 M. Scan rate = 20 mV/s.

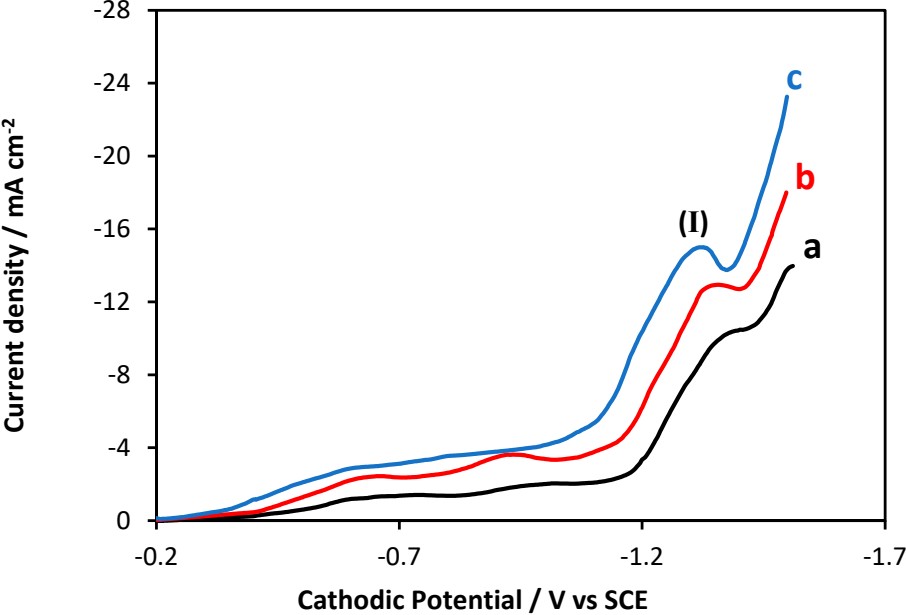

**Figure 4.** Cathodic polarization curves for the electrodeposition of Co–Cu alloy onto steel cathodes at pH = 10 from solutions containing 0.12 M $CoSO_4.7H_2O$, 0.03 M $CuSO_4.5H_2O$, 0.5 M lactic acid, and 0.1 M $Na_2SO_4$ at different temperatures: (**a**) 25; (**b**) 45; (**c**) 55 °C. Scan rate = 20 mV/s.

The composition of the deposit (w), deposition time (t), total mass of the deposit (m), and electrochemical equivalent (e) (mass of the deposited substance by passing 1 Ampere current for 1 s) can be used to calculate the partial current densities of individual metals ($i_p$) during codeposition [55,66].

$$i_p = \frac{mw}{100 \, et} \tag{16}$$

The current efficiency of hydrogen was calculated by collecting the hydrogen gas evolved during the electrodeposition process and measuring its volume. The volume of hydrogen gas was converted into mass using the ideal gas law, and the hydrogen evolution efficiency can be calculated as follows:

$$\text{Hydrogen evolution efficiency} = \frac{\text{mass of collected hydrogen gas}}{\text{theoretical mass of hydrogen}} \times 100 \qquad (17)$$

The cathodic polarization curve for Co–Cu alloy deposition is an aggregate curve representing the whole set of processes occurring at the cathode. The procedure for obtaining the true current–potential relations of metals in codeposition consists of resolving the curve of alloy deposition into two or more component curves of partial current density, one for each parent metal and one for hydrogen if it is also discharged. To do this, the composition of the alloy must be known over the range of the current density involved. The computed partial polarization curves of Cu, Co, and $H_2$ are also plotted in Figure 1 (the dashed curves). The calculated partial polarization curves differ greatly from the corresponding experimental polarization curves for individual deposition. The computed curve for Co ($i_{Co}$) is closer to that of alloy indicating that a high Co deposit is to be expected. This is in fair agreement with the results obtained.

The partial current density of Co is higher than that of Cu and improves with increasing applied current density. Because of the low $i_p$ value of hydrogen, alloys with high current efficiency were obtained. Since the percentage of Co in the deposit is smaller than its percentage in the bath, the electrodeposition of the Co–Cu alloy from the lactate bath follows the regular-type.

The cathodic polarization of Co–Cu alloys was affected by the $[Co^{2+}]/[Cu^{2+}]$ molar ratio (2–6.5), as shown in Figure 2. The polarization curves were affected when the $[Co^{2+}]/[Cu^{2+}]$ molar ratio in the bath decreased, moving in the direction of higher positive values, i.e., the polarization curves shift to more noble values by either decreasing the concentration of $Co^{2+}$ ions or increasing the concentration of $Cu^{2+}$ ions. Because of more $Cu^{2+}$ ions in the cathodic diffusion layer, the cathodic (concentration) polarization is reduced. Figure 3 depicts how the concentration of lactic acid affects the cathodic polarization of the Co–Cu alloy. As the concentration of lactic acid increases in the bath from 0.5 to 0.8 M, the deposition potential of the Co–Cu alloy shifts from −1.2 to −1.15 V. Lactic acid functions as a catalyst for the reduction of $Co^{2+}$ ions [51].

The cathodic polarization of the Co–Cu alloy deposition as a function of temperature is shown in Figure 4. The polarization curves shift towards more positive values as the bath temperature rises from 25 to 55 °C because of a decrease in the activation type of overvoltage. Moreover, as the temperature rises, the rate at which the ionic species ($Co^{2+}$, $Cu^{2+}$, and $H^+$) diffuse into the cathodic diffusion layer also increases. This reduces the concentration type of polarization [23]. Each polarization curve shows a distinct peak (I) at −1.3 V. The deposition process was controlled by the diffusion process because the current density declines after the cathodic peak (I) [67,68].

*3.3. Composition of Co–Cu Electrodeposited Alloy*

According to the needs of the application, there are different Co–Cu compositions that work best. Co–Cu alloys with a cobalt content of 20 to 40% are reported as the most suitable and adaptable. This can be attributed to the alloys' excellent balance of strength, ductility, wear resistance, and hardness within this composition range [69].

A high CCE minimizes energy waste and lowers production costs by ensuring that a sizable amount of the electrical energy supplied to the electrochemical process is used for the intended metal or alloy deposition. A high CCE lessens waste and byproduct creation, decreasing the electrodeposition process's negative environmental effects [50].

Figures 5–8 show plots of the binary Co–Cu alloy composition and CCE as a function of different plating factors. The cathodic current efficiency of alloy deposition is the sum of the separate cathodic efficiencies of deposition of Co and Cu in that alloy. Because of the

concurrent reduction in hydrogen ions, the CCE for the deposition of Co–Cu alloys is high but less than 100%. The composition curves of Co lie below the composition reference line (CRL). The latter represents the percentage of Co in the bath. These findings suggest that the deposition of Cu was preferred over that of Co. This behavior is typical of plating in regular systems [70].

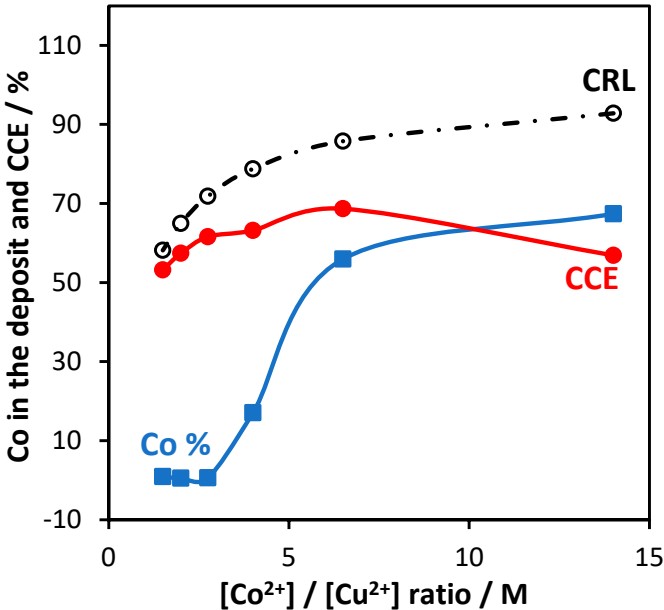

**Figure 5.** The effect of $[Co^{2+}]/[Cu^{2+}]$ molar ratio on CCE and the percentage of Co in the deposit from a bath containing 0.5 M lactic acid and 0.1 M $Na_2SO_4$ at pH = 10, c.d. = 3.33 mA cm$^{-2}$, time = 20 min., and temp. = 25 °C. The composition reference line (CRL) represents the percentage of Co in the bath.

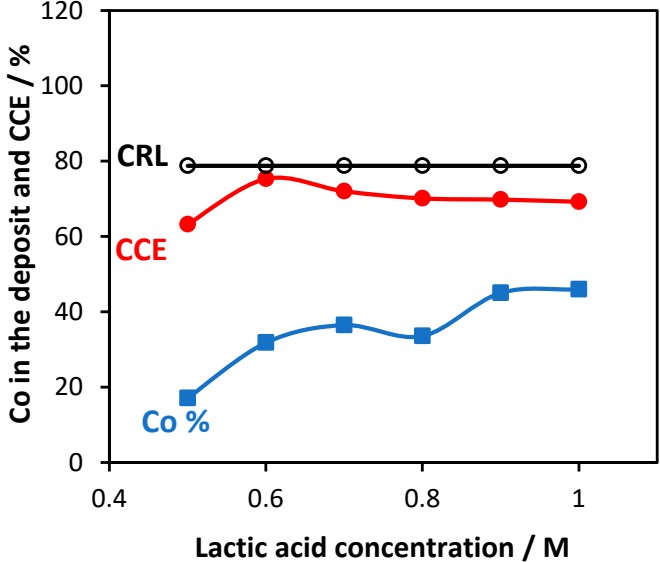

**Figure 6.** The effect of lactic acid concentration on CCE and the percentage of Co in the deposit from a bath containing 0.12 M CoSO$_4$.7H$_2$O, 0.03 M CuSO$_4$.5H$_2$O, and 0.1 M Na$_2$SO$_4$ at pH = 10, c.d. = 3.33 mA cm$^{-2}$, time = 20 min., and temp. = 25 °C. The composition reference line (CRL) represents the percentage of Co in the bath.

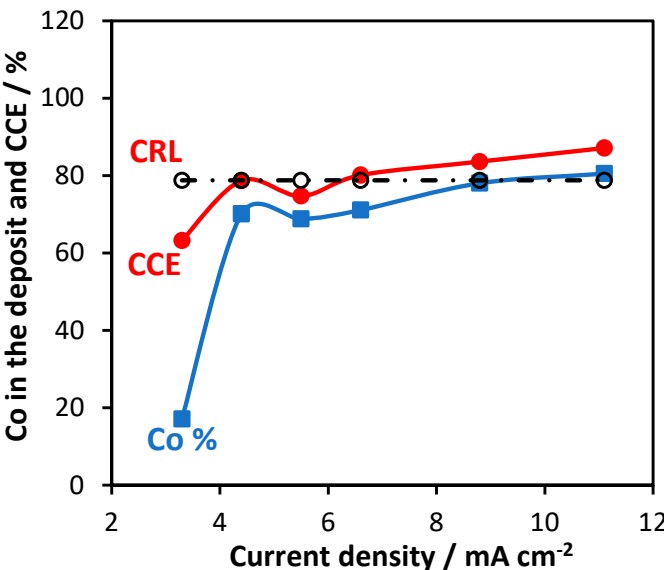

**Figure 7.** The effect of current density on CCE and the percentage of Co in the deposit from a bath containing 0.12 M CoSO$_4$.7H$_2$O, 0.03 M CuSO$_4$.5H$_2$O, 0.5 M lactic acid, and 0.1 M Na$_2$SO$_4$ at pH = 10, time = 20 min., and temp. = 25 °C. The composition reference line (CRL) represents the percentage of Co in the bath.

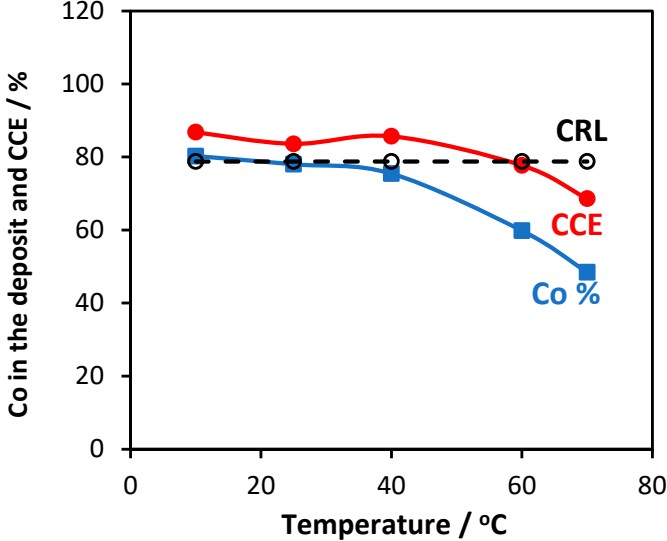

**Figure 8.** The effect of temperature on CCE and the percentage of Co in the deposits from a bath containing 0.12 M CoSO$_4$.7H$_2$O, 0.03 M CuSO$_4$.5H$_2$O, 0.5 M lactic acid, and 0.1 M Na$_2$SO$_4$ at pH = 10, c.d. = 8.89 mA cm$^{-2}$, and time = 20 min. The composition reference line (CRL) represents the percentage of Co in the bath.

The influence of the [Co$^{2+}$]/[Cu$^{2+}$] molar ratio (1.5–14) on the Co content of the deposited alloy and the CCE is depicted in Figure 5. As the [Co$^{2+}$]/[Cu$^{2+}$] molar ratio increases from 1.5 to 14, the Co% in the coating noticeably increases from 1 to 67%. At molar ratio 1.5, the cobalt deposition completely stops, leaving only copper in the deposit. This greatly improves the deposition efficiency of copper. The rate of Co reduction increases as the concentration of cobalt in the bath increases, which compensates for the loss of Co$^{2+}$ ions in the diffusion layer [71]. When the [Co$^{2+}$]/[Cu$^{2+}$] molar ratio rises from 1.5 to 6.5, the CCE of alloy deposition improves from 52 to 68%. As the deposit falls off the steel surface above this value (6.5), the CCE of the alloy deposition decreases to 51% at a molar ratio of 14. The percentage of Co in the deposit increases from 18 to 45% as the

concentration of lactic acid increases from 0.5 to 1 M. (Figure 6). The addition of lactic acid facilitates Co deposition by complexation and creating a homogeneous catalyst. Abd El-Rehim et al. [42] reported similar results with citrate ions. In addition, lactate ions can function as a catalyst for Co–Cu alloy discharge across the steel cathode without preventing alloy discharge overactive sites on the steel surface [72]. The CCE of alloy deposition rises to 78%, then tends to level off at 74% by increasing the lactic acid concentration from 0.5 to 1 M (Figure 6). The impact of current density in the range of 3.33–11.55 mA cm$^{-2}$ on the alloy composition and CCE is shown in Figure 7. As the current density increases, the cathodic polarization increases. This aids in the discharge of Co$^{2+}$ ions and improves the CCE of the codeposition. The deposit's cobalt content rises with increasing current density. Because Cu is deposited preferentially, the cathode surface is significantly more diminished by Cu$^{2+}$ ions than by Co$^{2+}$ ions. Therefore, when the current density increases, the deposit's Co% is likely to increase, whereas the Cu% tends to decrease. The quantity of the less noble metal (Co) increases to 80% in the deposit as the current density increases to 11.55 mA cm$^{-2}$. This is a normal trend for deposition in the regular system [23]. Due to the significant improvement in Co deposition efficiency, the CCE increases to 85% at the current density of 11.55 mA cm$^{-2}$.

The composition and CCE of the Co–Cu alloy deposition were both significantly reduced when the temperature was elevated from 10 to 70 °C (Figure 8). An increase in temperature tends to increase the number of discharged ions, Cu$^{2+}$ and Co$^{2+}$, in the cathode diffusion layer. This facilitates the deposition of Cu metal at the expense of Co. The Co content of the deposited alloy is reduced to 48.5% as the temperature increases to 70 °C. For an alloy deposited using a regular plating system, this behavior is normal. The CCE of alloy deposition is a constant value (85%) up to 40 °C. After that, it decreases to 65% as the temperature rises to 70 °C because of hydrogen evolution.

### 3.4. Morphology and Structure of Co–Cu Alloy Coatings

The formation of complexes in solution has a pronounced effect on metals' equilibrium potentials, the extent of overpotential, and the types of deposits that are formed cathodically. The overpotential often tends to increase, and the deposit's crystal size decreases. Additionally, there is a decrease in the tendency to create and develop dendrites [49]. All the Co–Cu alloy plates that were prepared are, in general, attached to the steel substrate. With increasing concentrations of Cu$^{2+}$ ions and lactic acid in the bath, the deposit acquires a more metallic appearance and brightness. The coatings' brightness is reduced as the current density or temperature is increased. According to the findings of the EDX study, copper and cobalt are the two main components of all the prepared alloy coatings.

Hydrogen evolution is a common side reaction that occurs during the electrodeposition of metals and alloys. The hydrogen bubbles can interfere with the growth of the metal deposit, leading to a variety of morphological changes, such as increasing porosity, decreasing adhesion, and increasing the roughness of the surface with dendritic or nodular morphology. The influence of the current density on the morphology and microstructure of the electrodeposited Co–Cu alloy is shown in Figure 9. The EDX spectrum shows that at a low current density (3.33 mA cm$^{-2}$), Cu metal is the main constituent with a weight percent of 82.9%, while cobalt is 17.1%. This mainly occurs because the applied current density is smaller than the copper-limiting current density. The obtained deposits are compact, coarse-grained, and dense (Figure 9a). Moreover, at a high current density (11.11 mA cm$^{-2}$), needle-shaped crystallites are observed, as well as larger ones with a spheroidal shape (Figure 9b). The bubbles of hydrogen gas cause the deposit to become rougher with a more dendritic or nodular morphology [73]. This shape might be ascribed to the deposit's increase in Co% as the current density increases.

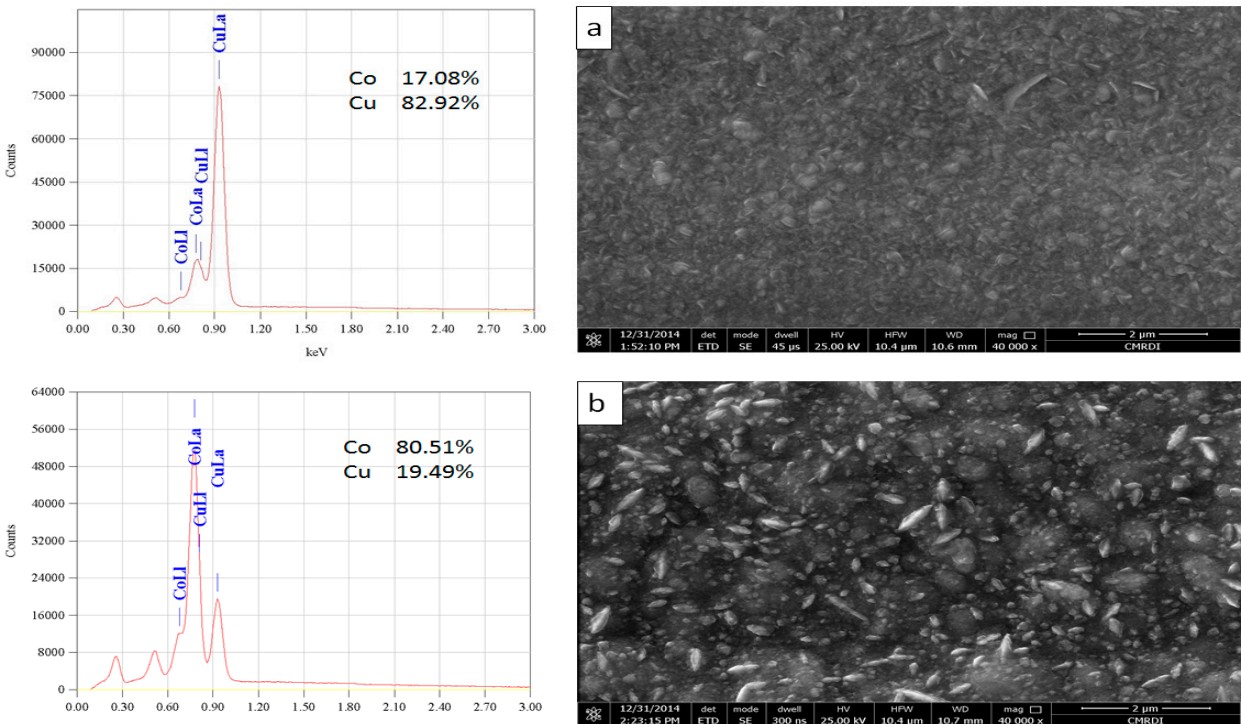

**Figure 9.** SEM images and EDX analysis of electrodeposited Co–Cu alloy obtained from a bath containing 0.12 M CoSO$_4$.7H$_2$O, 0.03 M CuSO$_4$.5H$_2$O, 0.5 M lactic acid, and 0.1 M Na$_2$SO$_4$ at pH = 10, time = 20 min., temp. = 25 °C, and at different current densities: (**a**); (**b**) 11.11 mA cm$^{-2}$.

The morphology and microstructure of the electrodeposited Co–Cu alloy are shown in micrographs of Figure 10, in relation to bath temperature. The impact of low temperature (10 °C) is depicted in Figure 10a. The deposits have small grains that are flat, compact, and smooth. When the bath temperature is increased to 70 °C (Figure 10b), homogeneous deposits with large grains are formed. Cathodic polarization is reduced as the temperature increases, favoring the growth rate mechanism over the nucleation rate mechanism.

One of the most important characteristics of Co–Cu alloy films that greatly affects their overall performance and applicability is their crystallinity. When compared to their amorphous or disordered equivalents, crystalline Co–Cu alloy films have better mechanical, electrical, and thermal stability due to their well-defined atomic structure. For electrodeposited metals and alloys, there is no single acceptable grain size. The optimum grain size is determined by the desired qualities and particular application. Gathering deposits with grains smaller than 100 nanometers are becoming more and more popular these days to create novel, innovative substances [74].

Co–Cu alloys electrodeposited from selected electrolytes under different practical settings were subjected to XRD analysis. The data show that the electrodeposited Co–Cu alloy exhibits a polycrystalline structure with a combination of copper-rich, face-centered cubic (FCC) Co–Cu and cobalt hexagonal close-packed (HCP) phases. The presence of the two phases indicates that the alloy composition lies within the two-phase region of the Co–Cu phase diagram [70]. Cu–Co alloy coatings produced by employing citrate, glycine, or tartrate baths showed the same results [17,18,26].

The strong and sharp peaks at 44.3°, 51.6°, and 74.5° 2θ correspond to the (111), (200), and (220) reflections, respectively, of the Co–Cu with FCC structure. The weaker and broader peaks at around 41.5°, 47°, and 76.3° 2θ are attributed to the (100), (101), and (110) reflections of the HCP phase. This indicates a significant amount of cobalt in the alloy. The presence of both phases implies that the alloy exhibits a combination of property characteristic of both Cu and Co.

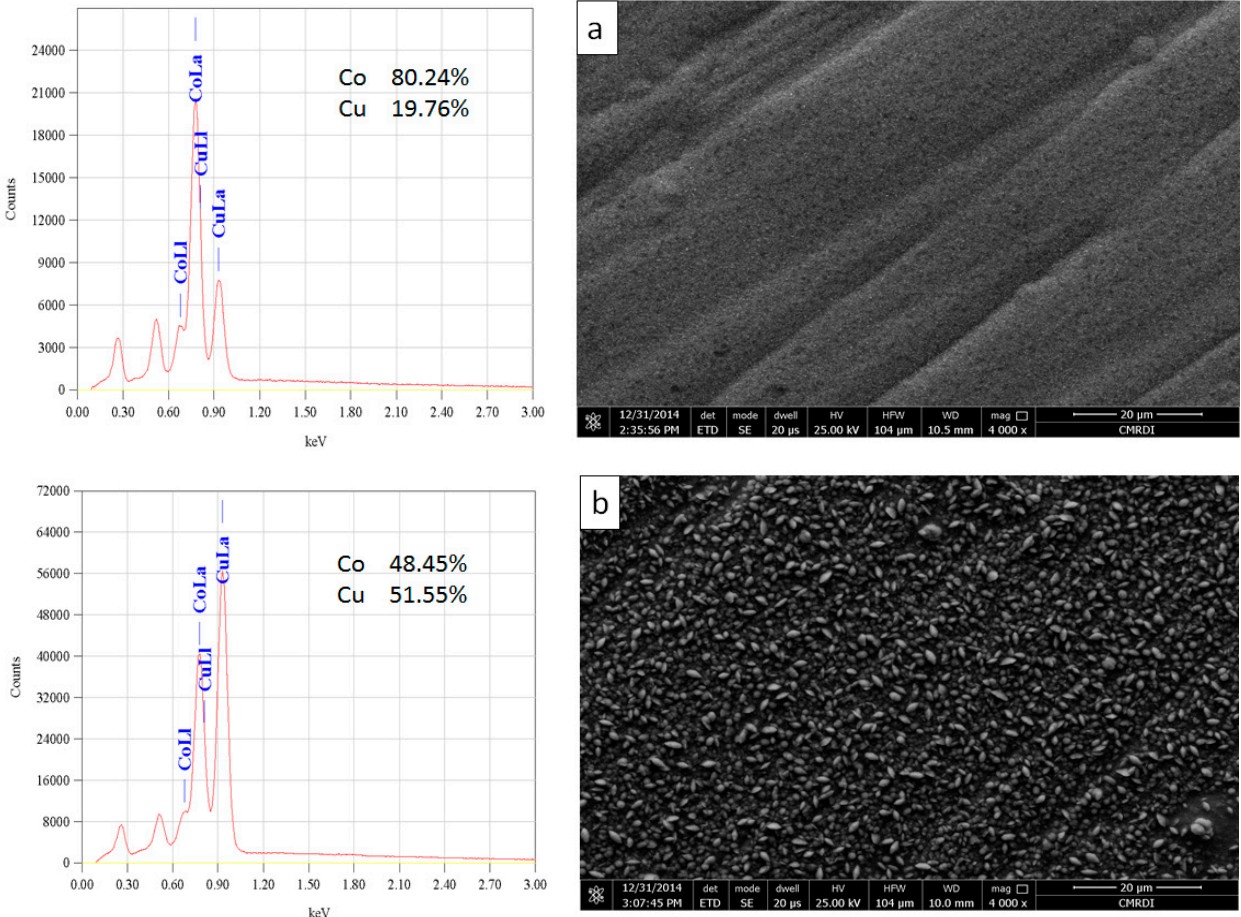

**Figure 10.** SEM images and EDX analysis of electrodeposited Co–Cu alloy obtained from a bath containing 0.12 M CoSO$_4$.7H$_2$O, 0.03 M CuSO$_4$.5H$_2$O, 0.5 M lactic acid, and 0.1 M Na$_2$SO$_4$ at pH = 10, time = 20 min., c.d. = 8.89 mA cm$^{-2}$, and at different temperatures: (**a**) 10 °C; (**b**) 70 °C.

The weight percentage of Cu in the coatings obtained in this work ranges from 19.8 to 99%. As has already been noted in the literature, it is, therefore, more likely that the coatings were made of a supersaturated metastable solid solution of Cu and Co [26]. The method employed, however, was unable to identify any segregated phases. These findings support the limitations of traditional XRD for examining the non-homogeneities in Cu–Co solid solutions, particularly in cases where thin films are produced.

Figure 11 shows the influence of current density on the XRD patterns of the electrodeposited Co–Cu alloy. The alloy of Cu$_{82.9}$ Co$_{17.1}$ composition exhibits three clearly diffraction peaks—(111), (200), and (220)—at a low current density (3.33 mA cm$^{-2}$) (Figure 11b). Sharp and defined peaks signify good crystallization. The average particle size of the crystallites, calculated using the Scherrer equation, is 89 nm. The Co–Cu alloy peaks' intensity decreases and becomes broad, and their positions shift towards higher 2θ values as the current density increases to 11.11 mA cm$^{-2}$ (Figure 11c). This is due to a decrease in the crystallite size (69 nm) and a change in the chemical composition of the deposited alloy (Cu$_{19.5}$ Co$_{80.5}$) at a higher current density. The lattice constants continuously alter as a function of composition, which causes peak-shifting [57]. There are two minor peaks of Co HCP that are matched with the ICDD # 00-001-1278 card. The peaks are attributed to (100) and (101) reflections. The Co metal peaks that arise at high current densities are in good agreement with the EDX data in Figure 9b. For comparison, the standard pattern (Figure 11a) with a composition of Cu$_{48}$ Co$_{52}$ is provided for the Co–Cu alloy [75].

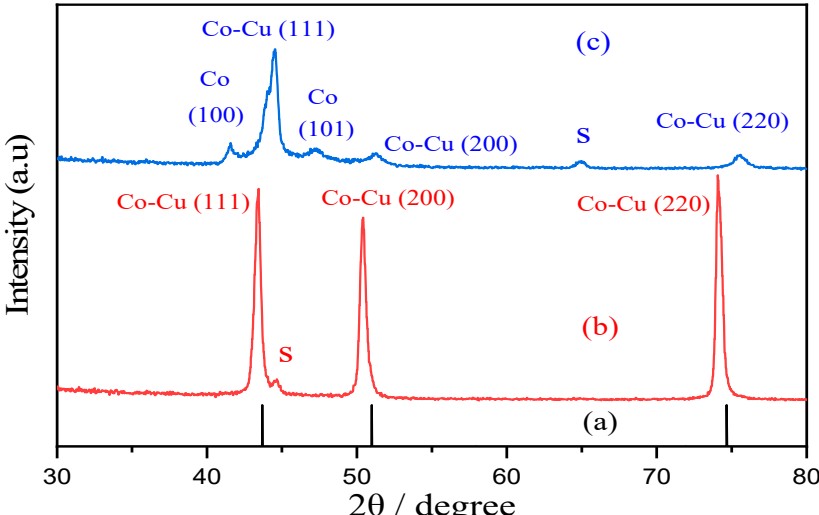

**Figure 11.** XRD patterns of electrodeposited Co–Cu alloy obtained from a bath containing 0.12 M $CoSO_4.7H_2O$, 0.03 M $CuSO_4.5H_2O$, 0.5 M lactic acid, and 0.1 M $Na_2SO_4$ at pH = 10, time = 20 min., temp. = 25 °C, and at different current densities: (**a**) ICDD 00-050-1452; (**b**) 3.33 mA $cm^{-2}$; (**c**) 11.11 mA $cm^{-2}$.

The impact of temperature (10–70 °C) on the structure of the electrodeposited Co–Cu alloy is shown in Figure 12. A strong (111) peak with high intensity and a minor (200) peak with low intensity, which is attributed to the FCC of the Co–Cu phase ($Cu_{19.8}$ $Co_{80.2}$), could be observed at a low temperature (Figure 12b). Additionally, there are three minor peaks—(100), (101), and (110)—that correspond to the Co HCP phase. The average crystallite size of the coating is 74 nm. It is worth observing that, for the electrodeposited alloy ($Cu_{51.6}$ $Co_{48.5}$) at 70 °C, the (111) peak splits into two peaks. This splitting could be attributed to the formation of a new phase at elevated temperatures (Figure 12c). At 70 °C, the average crystallite size is reduced to 46 nm. The XRD data are in good agreement with the EDX analysis.

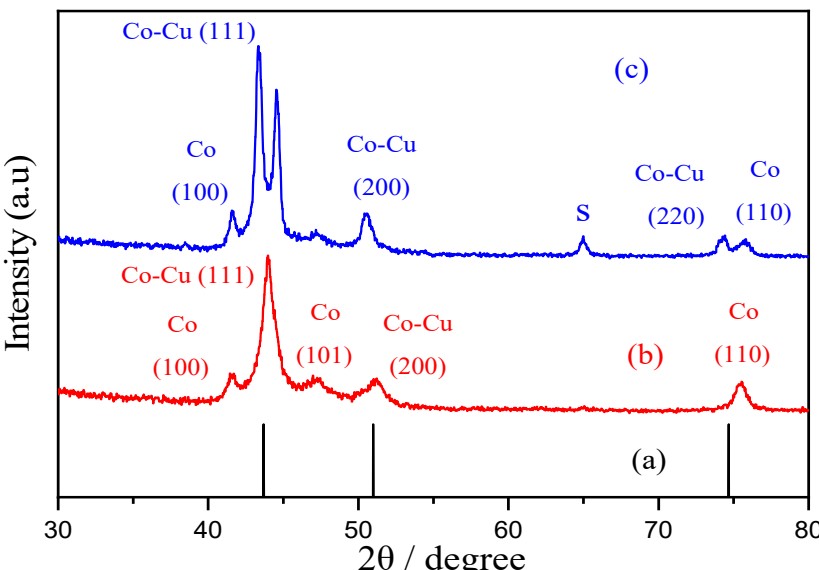

**Figure 12.** XRD patterns of electrodeposited Co–Cu alloy obtained from a bath containing 0.12 M $CuSO_4.7H_2O$, 0.03 M $CuSO_4.5H_2O$, 0.5 M lactic acid, and 0.1 M $Na_2SO_4$ at pH = 10, time = 20 min., c.d. = 8.89 mA $cm^{-2}$, and at different temperatures: (**a**) ICDD 00-050-1452, (**b**) 10 °C, (**c**) 70 °C.

### 3.5. T.P and T.I Measurements

Throwing power is an important electroplating characteristic, especially when trying to achieve uniform Co–Cu alloy film deposition. It describes an electroplating bath's capacity to uniformly distribute the metal ions that are deposited across the cathode surface, even in the presence of changes in current density. Strong throwing power in the plating bath guarantees uniform thickness of the Co–Cu alloy coating over the whole surface, avoiding the development of thinner deposits in recessed areas and thicker deposits near edges and corners [76].

The impacts of the composition of the bath and various operation conditions on the lactate bath's throwing power for the electroplating of Co–Cu alloys are investigated. In Figure 13a–d, a linear plot of the linear ratio (L) and the metal distribution ratio (M) is displayed. By comparing the slope's reciprocal (T.I), the baths' throwing power is evaluated. Table 1 lists the values of T.I. According to Figure 13a and Table 1, the T.P and T.I increase to 2 and 67.3%, respectively, as the $[Co^{2+}]/[Cu^{2+}]$ molar ratio decreases to 4. The magnitudes of the T.P and T.I decrease to 1.6 and 45%, respectively, upon increasing the concentration of lactic acid from 0.5 to 1 M, as shown in Figure 13b. The T.P, however, improves to 88% when increasing the current density to 11.1 mA cm$^{-2}$ (Figure 13c), which is caused by an increase in cathodic polarization. Increasing the bath temperature to 70 °C causes the bath's T.P and T.I to reduce to 1 and 45%, respectively, because of a decrease in the cathodic polarization (Figure 13d). Although rising temperatures improves electrolyte conductivity, it is possible to hypothesize that this effect is caused by the depolarizing effect of higher temperatures. According to this finding, the depolarization impact is more effective.

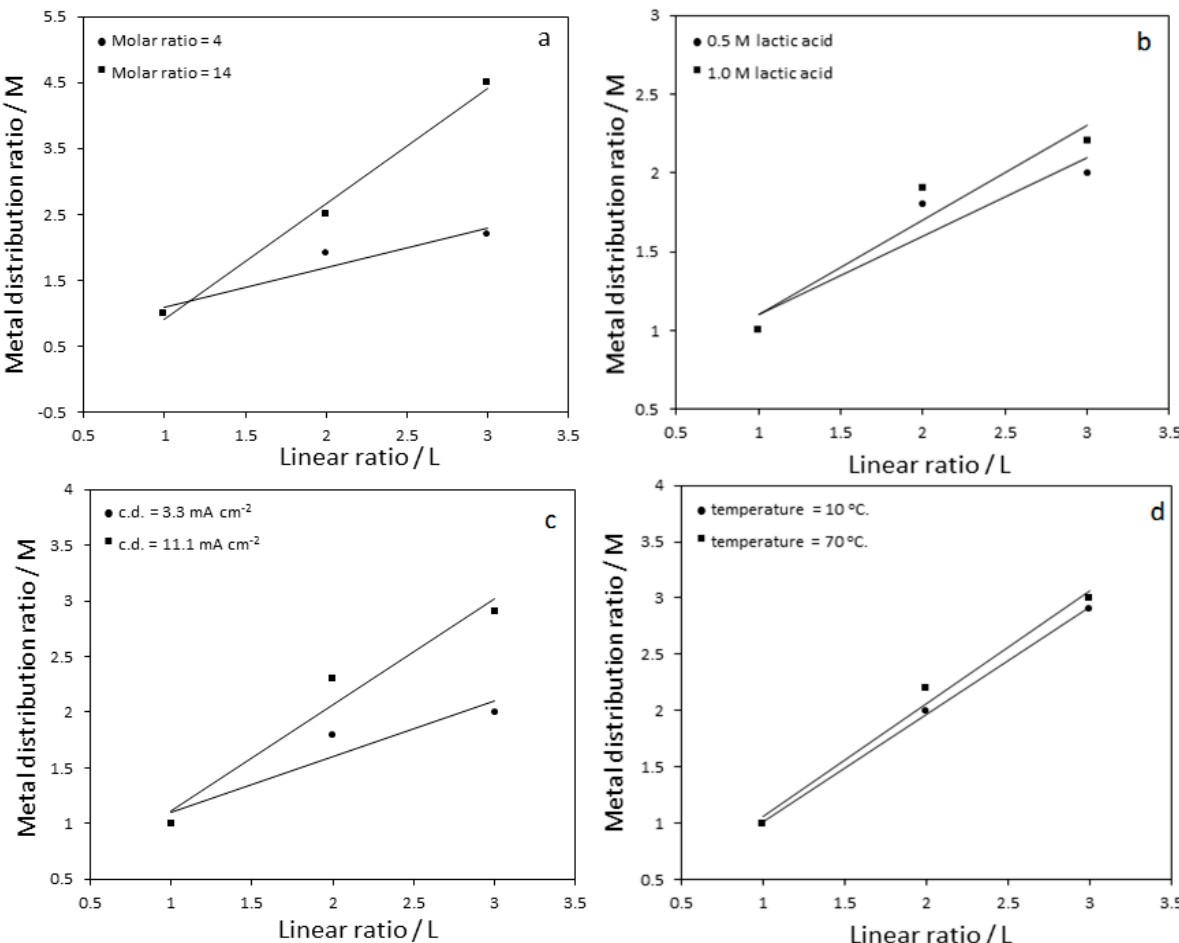

**Figure 13.** The effect of the $[Co^{2+}]/[Cu^{2+}]$ molar ratio (**a**), lactic acid concentration (**b**), current density (**c**), and temperature (**d**) on the T.I of an alkaline lactate bath.

**Table 1.** Effect of bath composition and operating conditions on the throwing power and throwing index of an alkaline lactate bath.

| Bath Composition | | | Operating Conditions | | | T.I | T.P % |
|---|---|---|---|---|---|---|---|
| Cobalt Sulphate (M) | Copper Sulphate (M) | Lactic Acid (M) | C.D (mA cm$^{-2}$) | Time (min.) | Temp. (°C) | | |
| 0.14 | 0.01 | 0.5 | 3.3 | 20 | 25 | 0.5 | 48 |
| 0.12 | 0.03 | 0.5 | 3.3 | 20 | 25 | 2.0 | 67.3 |
| 0.12 | 0.03 | 1.0 | 3.3 | 20 | 25 | 1.6 | 45 |
| 0.12 | 0.03 | 0.5 | 11.1 | 20 | 25 | 1.1 | 88 |
| 0.12 | 0.03 | 0.5 | 8.8 | 20 | 10 | 1.1 | 70 |
| 0.12 | 0.03 | 0.5 | 8.8 | 20 | 70 | 1.0 | 45 |

*3.6. Comparison of the Cathodic Current Efficiency of the Lactate Bath with the Previously Discussed Baths*

In Table 2, the cathodic current efficiency of the lactate bath under investigation is compared with that of other baths mentioned in the literature. For the Co–Cu deposits that are obtained from the glycinate bath [23], the efficiency of deposition was 84%. De Souza et al. [26] reported 65% efficiency for the deposition of Co–Cu alloy from a sodium tartrate bath. The obtained deposits were needle-form grains. The deposition from the citrate–boric bath attained 88% efficiency [35]. The addition of boric acid gave bright and smooth deposits with a metallic luster. A Co–Cu alloy with an efficiency of 85% was obtained from a sulphate bath by Mentar et al. [77]. Ignatova and Lilova [78] electrodeposited Co–Cu alloys from ammonia–sulphate and citrate electrolytes with an efficiency of 90%. The deposits obtained from the ammonia–sulphate electrolyte had a coarse-grained structure with an average size of 2–10 μm. However, needle-shaped crystals, as well as larger ones with a spheroidal shape, were obtained from citrate electrolyte. Lei et al. [79] deposited a Co–Cu alloy from a bath containing boric acid and saccharine. The deposition efficiency ranged between 60 and 100%, and the obtained deposits were tiny crystals oriented in all directions. For the present work, the deposition efficiency of the Co–Cu alloy from the lactate bath was 85%, and the deposited alloy was smooth and compact with an average size of 46–89 nm. The comparison demonstrates that the lactate bath has good efficiency and produces satisfactory deposits.

**Table 2.** A comparison between the cathodic current efficiency (CCE) of the lactate bath and other baths reported in the literature for the electrodeposition of Co–Cu alloys.

| Alloy | Bath | CCE | Reference |
|---|---|---|---|
| Co–Cu | Glycinate | 84% | [23] |
| Co–Cu | Tartrate | 65% | [26] |
| Co–Cu | Citrate–boric | 88% | [42] |
| Co–Cu | Sulphate | 85% | [77] |
| Co–Cu | Ammonia-sulphate and citrate | 90% | [78] |
| Co–Cu | Boric–Saccharine | 60%–100% | [79] |
| Co–Cu | Lactate | 85% | This work |

**4. Conclusions**

The electrodeposition of Co–Cu thin alloy films onto steel substrates using a new alkaline lactate bath was studied under different experimental conditions. The kinetic effects have a significant contribution to enabling Co–Cu deposition. The deposition of Co–Cu alloy is of a regular type. The current efficiency for the alloy deposition is 85%. The efficiency of deposition is enhanced by increasing the lactic acid concentration, increasing the current density, and decreasing the temperature. The optimum conditions for producing satisfactory Co–Cu deposits were 0.12 M CoSO$_4$.7H$_2$O, 0.03 M CuSO$_4$.5H$_2$O, 0.5 M lactic acid, and 0.1 M Na$_2$SO$_4$ at pH = 10, c.d. = 8.89 mA cm$^{-2}$, and time = 20 min. The Co% in the

deposited alloy increases when increasing the $[Co^{2+}]/[Cu^{2+}]$ molar ratio, thereby increasing the lactic acid concentration, increasing the applied current density, and decreasing the temperature. The EDX analysis confirmed the presence of both Co and Cu metals in the coating alloy. The electrodeposited Co–Cu is composed of Co hexagonal close-packed (HCP) and Cu-rich face-centered cubic (FCC) Co–Cu phases, as confirmed via XRD. The size of the deposited crystallite is 46–89 nm. SEM confirmed the formation of dendritic and rougher deposits at a high current density. The alkaline lactate has good throwing power.

**Author Contributions:** Conceptualization, R.A.A. and M.M.K.; Methodology, M.M.M.; Software, M.M.M.; Validation, R.A.A. and M.M.K.; Investigation, M.M.K. and M.M.M.; Writing—original draft, M.M.K.; Writing—review & editing, M.M.K.; Project administration, R.A.A.; Funding acquisition, R.A.A. All authors have read and agreed to the published version of the manuscript.

**Funding:** This research was supported by Deanship of Scientific Research at Najran University under the General Research Funding program grant code (NU/DRP/SERC/12/29).

**Institutional Review Board Statement:** Not applicable.

**Informed Consent Statement:** Not applicable.

**Data Availability Statement:** Data will be made available on request.

**Acknowledgments:** The authors are thankful to the Deanship of Scientific Research at Najran University for funding this work under the General Research Funding program, grant code (NU/DRP/SERC/12/29).

**Conflicts of Interest:** The authors declare that they have no known competing financial interests or personal relationships that could have appeared to influence the work reported in this paper.

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
