# Peer review of "Electrodeposition of Nanostructured Co–Cu Thin Alloy Films on to Steel Substrate from an Environmentally Friendly Novel Lactate Bath under Different Operating Conditions"

_coatings, doi:10.3390/coatings14040407_

Round 1
Reviewer 1 Report
Comments and Suggestions for Authors
The work, as the name suggests, focuses on the Electrodeposition of nanostructured Co-Cu thin alloy films on to steel substrate from an environmentally friendly novel lactate bath under different operating conditions. The topic is relevant and the work has been carried out at a level that allows considering the paper for acceptance to the Coatings after responding to major comments.
1. There is no explanation why copper is predominantly deposited at low current densities (Fig. 4).
2. There is no reason to give the composition of the alloy with an accuracy of hundredths of a percent.
3. Why in the conclusions the optimal composition of the bath is given as “were 1.5-14 [Co2+]/[Cu2+] molar ratio” (line 462), and Fig. 4 and 5 are given for a different molar ratio, i.e. we don't see the "optimum".
4. Why do Fig. 1a show two lines (red and dotted) for the electrodeposition of copper from solution containing 0.03 M CuSO4.5H2O.
5. The authors do not explain the fact of obtaining alloys with a high cobalt content, because it is known that copper is easily deposited on steel without the participation of current.
6. The reactions for the production of lactates are given in eq. 8.9. According to publication Tianyu Chen et al 2019 J. Electrochem. Soc. 166 D761 DOI 10.1149/2.1231914jes, the stability constant copper(II)–lactate complexes is 10 5.05 Cu (H−1L) L−+OH− = Cu(H−1L)22−+H2O (for 7.80 < pH < 13.15).
7. The coordination number of Co (II) is 4. For this reason, the formula of the complex is [Co(C3H5O3)2(H2O)2] Journal of Synthetic Crystals 43(11):2971-2975 and 2986.
8. The experimental techniques need to be described in more detail. For example, it is not indicated how the EDX analysis was obtained: as an average value of several points or from one point. The brand of scales and scanning electron microscope are not indicated and etc.
9. All major sections are listed under the same number 1.
10. The caption to Figure 1 is given in excessive detail. All general descriptions for Figures 1a,b,c,d do not need to be repeated for each case. The working electrode is not provided. The name of the curves in figures 1 is the same as the numbering of the figures (a,b,c), which is confusing. Figure 1 itself should be placed in the place where it was first mentioned.
11. Figure 2 should contain not only the reference, but also permission to copy.
12. The images in Figure 8 are somewhat blurred.
13. The authors provide an extensive bibliography. Unfortunately, there are only 8 new references out of 62 (in the last 5 years), despite the relevance of the topic. It is recommended to add the novel refs, for example, a recent review (DOI 10.3390/met13040657) or the articles in the comments above.
14. The DOI index is not given for any refs.
Author Response
Dear Reviewer
Thank you very much for taking the time to review our manuscript entitled” Electrodeposition of nanostructured Co-Cu thin alloy films on to steel substrate from an environmentally friendly novel lactate bath under different operating conditions" for Journal of Coatings. We sincerely appreciate your insightful comments and suggestions, which have been very helpful in improving the quality of our work.
We have carefully considered each of your points and have revised the manuscript. We believe that these revisions have strengthened our manuscript and made it a more valuable contribution to the field.
We appreciate your continued consideration of our manuscript for publication in the journal of Coatings. Please do not hesitate to contact us if you have any further questions.
Sincerely,
Medhat M. Kamel
on behalf of the authors
Respond to the Comments:
Open Review
(x) I would not like to sign my review report
( ) I would like to sign my review report
Quality of English Language
(x) I am not qualified to assess the quality of English in this paper
( ) English very difficult to understand/incomprehensible
( ) Extensive editing of English language required
( ) Moderate editing of English language required
( ) Minor editing of English language required
( ) English language fine. No issues detected
Yes |
Can be improved |
Must be improved |
Not applicable |
|
1) Does the introduction provide sufficient background and include all relevant references? |
( ) |
(x) |
( ) |
( ) |
2) Are all the cited references relevant to the research? |
( ) |
(x) |
( ) |
( ) |
3) Is the research design appropriate? |
( ) |
( ) |
(x) |
( ) |
4) Are the methods adequately described? |
( ) |
( ) |
(x) |
( ) |
5) Are the results clearly presented? |
( ) |
( ) |
(x) |
( ) |
6) Are the conclusions supported by the results? |
( ) |
( ) |
(x) |
( ) |
Respond to the Comments
(1) The introduction part is improved, and new references are added that are related to the subject of the manuscript. The introduction part now contains 52 references.(2) The references are improved, and new recent references are added. (3) We improved the research design by replacing Fig.1 (a, b, c, d) with Fig. 1, Fig. 2, Fig. 3, and Fig. 4. Also, we removed Fig. 2, and improved Fig.8. We adjusted the type and the size of the text in the whole manuscript. I hope that the paper will be agreed with you. (4) The experimental methods are well revised and well described. See the materials and methods part in the text.(5) The results are rewritten again and clearly presented.(6) The conclusion part is improved and rewritten again.
Comments and Suggestions for Authors
The work, as the name suggests, focuses on the Electrodeposition of nanostructured Co-Cu thin alloy films on to steel substrate from an environmentally friendly novel lactate bath under different operating conditions. The topic is relevant, and the work has been carried out at a level that allows considering the paper for acceptance to the Coatings after responding to major comments.
- There is no explanation why copper is predominantly deposited at low current densities (Fig. 4).
- For the system employed here, Cu has a larger positive reversible potential than Co therefore, it is more noble than Co, and Cu is preferentially deposited. In addition, the applied current density is smaller than the copper limiting current density. See page 6, lines 232-245, please.
- There is no reason to give the composition of the alloy with an accuracy of hundredths of a percent.
- Thank you for your comment. The correction is done in the whole manuscript.
- Why in the conclusions the optimal composition of the bath is given as “were 1.5-14 [Co2+]/[Cu2+] molar ratio” (line 462), and Fig. 4 and 5 are given for a different molar ratio, i.e. we don't see the "optimum".
- The conclusion part is revised and rewritten again.
- When we studied the effect of various parameters, such as the molar ratio of cobalt / copper, the concentration of lactic acid, the current density, and temperature on the composition of the alloy and the current efficiency of deposition (Figs. 5-8), we used in the first parameter, which is the molar ratio of cobalt / copper, a wide range of concentration ratios ranging between 1.5 and 14. Then we selected the ratio that gives the highest current efficiency with a regular and good-shaped deposit. For the second factor, which is the effect of lactic acid concentration, we used the previous selected ratio, and then again, we selected the acid concentration that gives the highest current efficiency and a good-looking appearance. After that, we studied the effect of current density using the best molar ratio of cobalt / copper and the best concentration of lactic acid. Therefore, when studying the temperature effect, we already used the best molar ratio of cobalt / copper, the best concentration of acid, and the best current density. In other words, the composition of the plating bath and the conditions of electroplating that shown in the caption of figure 8 are the best / optimum for the deposition of the cobalt-copper alloy because they gave the highest current efficiency and the good appearance of the coating alloy.
- Why do Fig. 1a show two lines (red and dotted) for the electrodeposition of copper from solution containing 0.03 M CuSO4.5H2
- Both copper and cobalt have TWO lines (solid and dotted). The solid line is the experimental curve that obtained when the metal is deposited alone from the plating bath. The dotted line represents the partial current density of the metal (Co or Cu) during the electrodeposition of Co-Cu alloy. It can be calculated by using equation (16), page 8.
- The authors do not explain the fact of obtaining alloys with a high cobalt content, because it is known that copper is easily deposited on steel without the participation of current.
- For the system employed here, Cu has a larger positive reversible potential than Co therefore, it is more noble than Co, and Cu is preferentially deposited if at comparable bulk concentrations than Co. Getting a cobalt-rich alloy might be challenging due to the difference in nobility. To decrease the content of Cu in a Co alloy, a mass transport limit of Cu is created by lowering its bulk concentration relative to Co. To deposit copper, the applied current density must remain below the copper limiting current density. When the applied current density becomes bigger than the Cu limiting current density, or the working electrode potential is larger than the Co equilibrium potential, the less noble Co begins to be deposited. But at the same time copper is also deposited in the cobalt-layer. Because the copper concentration is low in the electrolyte, copper reduction is under transport control, and thus the copper content in the alloy is small and decreases with increasing the applied current density or decreasing temperature. The experimental conditions, such as electrolyte concentration, pH, and applied potential, can also influence the relative deposition rates of copper and cobalt.
- This discussion is inserted in the manuscript. See page 6, lines 232-245, please.
- The reactions for the production of lactates are given in eq. 8.9. According to publication Tianyu Chen et al 2019 J. Electrochem. Soc. 166 D761 DOI 10.1149/2.1231914jes, the stability constant copper(II)–lactate complexes is 10 05 Cu (H−1L) L−+OH−= Cu(H−1L)22−+H2O (for 7.80 < pH < 13.15).
- Thank you for the valuable comment. The correction is done, and the reference is inserted. See equations 10, 12, and Reference 60, please.
7.The coordination number of Co (II) is 4. For this reason, the formula of the complex is [Co(C3H5O3)2(H2O)2] Journal of Synthetic Crystals 43(11):2971-2975 and 2986.
- Thank you for the comment. The correction is done, and the reference is inserted. See equations 11,13, and Reference 61, please.
- The experimental techniques need to be described in more detail. For example, it is not indicated how the EDX analysis was obtained: as an average value of several points or from one point. The brand of scales and scanning electron microscope are not indicated etc.
- The experimental techniques are described in detail. I hope it will be agreed with you. Thank you for the comment.
- All major sections are listed under the same number 1.
- Thank you for the remark. The correction is done.
10.The caption to Figure 1 is given in excessive detail. All general descriptions for Figures 1a,b,c,d do not need to be repeated for each case. The working electrode is not provided. The name of the curves in figures 1 is the same as the numbering of the figures (a,b,c), which is confusing. Figure 1 itself should be placed in the place where it was first mentioned.
- Figure 1 (a, b, c, d) is replaced by Fig.1, Fig. 2, Fig. 3, and Fig. 4.
- The working electrode is inserted in each figure caption.
- The correction is done.
- Figure 2 should contain not only the reference, but also permission to copy.
- Figure 2 is deleted.
- The images in Figure 8 are somewhat blurred.
- The figure is redrawn again. I hope it will be agreed with you. The figure becomes Fig. 13.
- The authors provide an extensive bibliography. Unfortunately, there are only 8 new references out of 62 (in the last 5 years), despite the relevance of the topic. It is recommended to add the novel refs, for example, a recent review (DOI 10.3390/met13040657) or the articles in the comments above.
- 14 Novel references are added. The suggested references by you are inserted. The total number of references is 79.
14.The DOI index is not given for any refs.
- The DOI index is given for most of the references.
Reviewer 2 Report
Comments and Suggestions for Authors
1. The introduction section made a good contextualization of the subject of the manuscript. Just some improvements:
-Page 1, Line 30: “cobalt” can be immediately represented by its chemical symbol Co.
-Page 2, Line 75: “cobalt-copper” can be represented Co-Cu as along all manuscript.
-I believe that some parts of the introduction have a different letter type or size, i.e., text is not uniform along the introduction.
2. Line 77: “1. Materials and methods” replace to 2. Materials and methods.
3. Lines 78 and 79: “Electroplating of the Co-Cu alloy was conducted under constant current (galvanostatic) from a solution containing CoSO4.7H2O, CuSO4.5H2O, anhydrous Na2SO4, and lactic acid.” Please indicate the molar concentration of the solution and the constant current.
4. Line 100: a dot is missing at the end of the sentence “was used for polarization measurements”.
5. Line 121: “1. Results and discussion” replace to 3. Results and discussion.
6. Line 160 and 161: Please explain with higher detail the sentence: “Kinetic effects are believed to make an important contribution, enabling cobalt-copper codeposition”.
7. Along the manuscript some uniformity is required – use Co-Cu or cobalt-copper.
8. Section 3.2, the manuscript shows the cathodic polarization curves for the deposition of Co-Cu alloy (b, c and d) under different conditions, varying the molar ratios of Co2+/Cu2+, the concentration of lactic acid, and temperature. It would be important to introduce an explanation of the range of values tested for the different conditions.
9. Section 3.3, figure 2 is only an incorporation of literature work. Thus, explain the necessity of incorporating of this information here (results section).
10. Section 3.3, figure 3b, it appears as CR instead of CRL.
11. Lines 250-252: Please correct the sentence: “The composition reference line (CRL), which depicts the less noble metal percentage (Co) in the bath, lies below the composition curves of Co.”
12. Section 3.3, page7, in the discussion of the results, Figure 3b is wrongly changed by Figure 3c.
13. Was it performed some test with the best combination of all plating factors?
14. Section 3.4, it would be interesting to see in Figure 4 a sample (SEM and EDX) with an intermedium current density, between the 3.33 and 11.11 mAcm2. The same comment is applied to Figure 5, an intermedium temperature.
15. Figure 8 has a poor image quality.
16. In the study of the TP and TI with bath composition and operating conditions table 1 is a good summary of the results. Why presenting Figure 8?
17. Some parts of the conclusion have a different letter type or size, i.e., text is not uniform along the conclusion.
Author Response
Dear Reviewer
Thank you very much for taking the time to review our manuscript entitled” Electrodeposition of nanostructured Co-Cu thin alloy films on to steel substrate from an environmentally friendly novel lactate bath under different operating conditions" for Journal of Coatings. We sincerely appreciate your insightful comments and suggestions, which have been very helpful in improving the quality of our work.
We have carefully considered each of your points and have revised the manuscript. We believe that these revisions have strengthened our manuscript and made it a more valuable contribution to the field.
We appreciate your continued consideration of our manuscript for publication in the journal of Coatings. Please do not hesitate to contact us if you have any further questions.
Sincerely,
Medhat M. Kamel
on behalf of the authors
-------------------------------------
Open Review
(x) I would not like to sign my review report
( ) I would like to sign my review report
Quality of English Language
( ) I am not qualified to assess the quality of English in this paper
( ) English very difficult to understand/incomprehensible
( ) Extensive editing of English language required
( ) Moderate editing of English language required
( ) Minor editing of English language required
(x) English language fine. No issues detected
Yes |
Can be improved |
Must be improved |
Not applicable |
|
1) Does the introduction provide sufficient background and include all relevant references? |
(x) |
( ) |
( ) |
( ) |
2) Are all the cited references relevant to the research? |
(x) |
( ) |
( ) |
( ) |
3) Is the research design appropriate? |
( ) |
(x) |
( ) |
( ) |
4) Are the methods adequately described? |
( ) |
( ) |
(x) |
( ) |
5) Are the results clearly presented? |
( ) |
( ) |
(x) |
( ) |
6) Are the conclusions supported by the results? |
( ) |
(x) |
( ) |
( ) |
Respond to the Comments
(1) Thank you for your comment.(2) Thank you for your comment.(3) We improved the research design by replacing Fig.1 (a, b, c, d) with Fig. 1, Fig. 2, Fig. 3, and Fig. 4. Also, we removed Fig. 2, and improved Fig.8. We adjusted the type and the size of the text. I hope that the paper will be agreed with you. (4) The experimental methods are well revised and well described. See the materials and methods part in the text.(5) The results are rewritten again and clearly presented.(6) The conclusion part is improved and rewritten again.
Comments and Suggestions for Authors
- The introduction section made a good contextualization of the subject of the manuscript. Just some improvements:
-Page 1, Line 30: “cobalt” can be immediately represented by its chemical symbol Co.
- The correction is done in the whole manuscript.
-Page 2, Line 75: “cobalt-copper” can be represented Co-Cu as along all manuscript.
- The correction is done in the whole manuscript.
- I believe that some parts of the introduction have a different letter type or size, i.e., text is not uniform along the introduction.
- Thank you for your remark. A revision is done.
- Line 77: “1. Materials and methods” replace to 2. Materials and methods.
- The correction is done.
- Lines 78 and 79: “Electroplating of the Co-Cu alloy was conducted under constant current (galvanostatic) from a solution containing CoSO4.7H2O, CuSO4.5H2O, anhydrous Na2SO4, and lactic acid.” Please indicate the molar concentration of the solution and the constant current.
- The molar concentration of bath components and applied currents are inserted in the text.
- Line 100: a dot is missing at the end of the sentence “was used for polarization measurements”.
- Thank you very much for the comment. The dot is added.
- Line 121: “1. Results and discussion” replace to 3. Results and discussion.
- The correction is done.
- Line 160 and 161: Please explain with higher detail the sentence: “Kinetic effects are believed to make an important contribution, enabling cobalt-copper codeposition”.
- For the system employed here, Cu has a larger positive reversible potential than Co therefore, it is more noble than Co, and Cu is preferentially deposited if at comparable bulk concentrations than Co. Getting a cobalt-rich alloy might be challenging due to the difference in nobility. To decrease the content of Cu in a Co alloy, a mass transport limit of Cu is created by lowering its bulk concentration relative to Co. To deposit copper, the applied current density must remain below the copper limiting current density. When the applied current density becomes bigger than the Cu limiting current density, or the working electrode potential is larger than the Co equilibrium potential, the less noble Co begins to be deposited. But at the same time copper is also deposited in the cobalt-layer. Because the copper concentration is low in the electrolyte, copper reduction is under transport control, and thus the copper content in the alloy is small and decreases with increasing the applied current density or decreasing temperature. The experimental conditions, such as electrolyte concentration, pH, and applied potential, can also influence the relative deposition rates of copper and cobalt.
- This discussion is inserted in the manuscript.
- Along the manuscript some uniformity is required – use Co-Cu or cobalt-copper.
- The correction is done in the whole manuscript, and we used Co-Cu.
- Section 3.2, the manuscript shows the cathodic polarization curves for the deposition of Co-Cu alloy (b, c, and d) under different conditions, varying the molar ratios of Co2+/Cu2+, the concentration of lactic acid, and temperature. It would be important to introduce an explanation of the range of values tested for the different conditions.
- When choosing the values of the different variables used in this study, we considered that they were as follows: We used a wide range of cobalt /copper molar ratios ranging from 1.5 to 14, so that we can get both cobalt-rich alloys and copper-rich alloys. We used concentrations of lactic acid ranging from 0.5 to 1 M to increase the stability of the formed complexes between lactate ligand and cobalt and copper and thus obtain deposited alloy with different proportions of copper and cobalt. We also used a range of current density between 3.33 and 11.11 mA cm-2 so that we can get copper-rich alloys at low current densities and cobalt-rich alloys at high current densities. This has been confirmed by energy-dispersing X-ray analysis (Figs. 9 &10). Temperatures were used in the range of 10 to 70 oC to get copper-rich alloys at high temperatures and cobalt-rich alloys at low temperatures. This is again confirmed by the energy-dispersing X-rays analysis (Figs. 11 & 12).
- Section 3.3, figure 2 is only an incorporation of literature work. Thus, explain the necessity of incorporating of this information here (results section).
- The figure is deleted.
- Section 3.3, figure 3b, it appears as CR instead of CRL.
- Thank you for your comment. The correction is done. The mentioned figure becomes Fig. 7
- Lines 250-252: Please correct the sentence: “The composition reference line (CRL), which depicts the less noble metal percentage (Co) in the bath, lies below the composition curves of Co.”
- The correction is done (line 365-366). Thank you, sir.
- Section 3.3, page7, in the discussion of the results, Figure 3b is wrongly changed by Figure 3c.
- Thank you for your comment. The correction is done
- Was it performed some test with the best combination of all plating factors?
- When we studied the effect of various parameters, such as the molar ratio of cobalt / copper, the concentration of lactic acid, the current density, and temperature on the composition of the alloy and the current efficiency of deposition (Figs. 5-8), we used in the first parameter, which is the molar ratio of cobalt / copper, a wide range of concentration ratios ranging between 1.5 and 14. Then we selected the ratio that gives the highest current efficiency with a regular and good-shaped deposit. For the second factor, which is the effect of lactic acid concentration, we used the previous selected ratio, and then again, we selected the acid concentration that gives the highest current efficiency and a good-looking appearance. After that, we studied the effect of current density using the best molar ratio of cobalt / copper and the best concentration of lactic acid. Therefore, when studying the temperature effect, we already used the best molar ratio of cobalt / copper, the best concentration of acid, and the best current density. In other words, the composition of the plating bath and the conditions of electroplating that shown in the caption of figure 8 are the best / optimum for the deposition of the cobalt-copper alloy because they gave the highest current efficiency and the good appearance of the coating alloy.
- Section 3.4, it would be interesting to see in Figure 4 a sample (SEM and EDX) with an intermedium current density, between the 3.33 and 11.11 mAcm2. The same comment is applied to Figure 5, an intermedium temperature.
- I am very sorry, Your Honor, because I am currently away from my laboratory, which makes it very hard nowadays to make these analyses. Please accept my apology.
- Figure 8 has a poor image quality.
- The figure is redrawn again. I hope it will be agreed with you. The figure becomes Fig. 13.
- In the study of the TP and TI with bath composition and operating conditions table 1 is a good summary of the results. Why presenting Figure 8?
- Analyzing the (M)vs. (L) plot allows us to determine the throwing index (TI) and assess the bath's ability to deposit metal uniformly. By plotting M vs. L, we can visualize the relationship between the desired metal distribution (M) and the actual current distribution (L) in the bath. A steeper slope in the M vs. L plot indicates a larger difference between the two, signifying poor throwing index. A flatter slope in the M vs. L plot refers to a higher throwing index. This indicates the metal deposition is closer to the ideal scenario, where the metal distribution (M) closely follows the geometric current distribution (L) even in areas with lower current density.
- Some parts of the conclusion have a different letter type or size, i.e., text is not uniform along the conclusion.
- Thank you very much. The correction is done.
Round 2
Reviewer 1 Report
Comments and Suggestions for Authors
After a lot of work on responding to comments and correcting the manuscript, the paper is now fully ready for acceptance to the Coatings.
Reviewer 2 Report
Comments and Suggestions for Authors
The authors reply to all my comments.